# Long-Term Therapeutic Effects of ^225^Ac-DOTA-E[c(RGDfK)]_2_ Induced by Radiosensitization via G2/M Arrest in Pancreatic Ductal Adenocarcinoma

**DOI:** 10.3390/pharmaceutics17010009

**Published:** 2024-12-24

**Authors:** Mitsuyoshi Yoshimoto, Kohshin Washiyama, Kazunobu Ohnuki, Ayano Doi, Miki Inokuchi, Motohiro Kojima, Brian W. Miller, Yukie Yoshii, Anri Inaki, Hirofumi Fujii

**Affiliations:** 1Division of Functional Imaging, Exploratory Oncology Research & Clinical Trial Center, National Cancer Center, Kashiwa 277-8577, Japan; 2Advanced Clinical Research Center, Fukushima Global Medical Science Center, Fukushima Medical University, Fukushima 960-1295, Japan; kwashi@fmu.ac.jp; 3Division of Pathology, Exploratory Oncology Research & Clinical Trial Center, National Cancer Center, Kashiwa 277-8577, Japan; 4Department of Radiation Oncology, College of Medicine, University of Arizona, Tucson, AZ 85719, USA; 5National Institute of Radiological Sciences, National Institutes for Quantum Science and Technology, Chiba 263-8555, Japan

**Keywords:** radionuclide therapy, ^225^Ac, RGD peptide, pancreatic cancer, integrin, DNA damage

## Abstract

**Background**: Alpha radionuclide therapy has emerged as a promising novel strategy for cancer treatment; however, the therapeutic potential of ^225^Ac-labeled peptides in pancreatic cancer remains uninvestigated. **Methods**: In the cytotoxicity study, tumor cells were incubated with ^225^Ac-DOTA-RGD_2_. DNA damage responses (γH2AX and 53BP1) were detected using flowcytometry or immunohistochemistry analysis. Biodistribution and therapeutic studies were carried out in BxPC-3-bearing mice. **Results**: ^225^Ac-DOTA-RGD_2_ demonstrated potent cytotoxicity against cells expressing α_v_β_3_ or α_v_β_6_ integrins and induced G2/M arrest and γH2AX expression as a marker of double-stranded DNA breaks. ^225^Ac-DOTA-RGD_2_ (20, 40, 65, or 90 kBq) showed favorable pharmacokinetics and remarkable tumor growth inhibition without severe side effects in the BxPC-3 mouse model. In vitro studies revealed that 5 and 10 kBq/mL of ^225^Ac-DOTA-RGD_2_ swiftly induced G2/M arrest and elevated γH2AX expression. Furthermore, to clarify the mechanism of successful tumor growth inhibition for a long duration in vivo, we investigated whether short-term high radiation exposure enhances radiation sensitivity. Initially, a 4 h induction treatment with 5 and 10 kBq/mL of ^225^Ac-DOTA-RGD_2_ enhanced both cytotoxicity and γH2AX expression with 0.5 kBq/mL of ^225^Ac-DOTA-RGD_2_ compared to a treatment with only 0.5 kBq/mL of ^225^Ac-DOTA-RGD_2_. Meanwhile, the γH2AX expression induced by 5 or 10 kBq/mL of ^225^Ac-DOTA-RGD_2_ alone decreased over time. **Conclusions**: These findings highlight the potential of using ^225^Ac-DOTA-RGD_2_ in the treatment of intractable pancreatic cancers, as its ability to induce G2/M cell cycle arrest enhances radiosensitization, resulting in notable growth inhibition.

## 1. Introduction

Pancreatic ductal adenocarcinoma (PDAC) is an aggressive disease, with a 5-year survival rate of <10%, and it is the third leading cause of cancer deaths in the USA. In addition, the mortality rate of PDAC is much higher than its incidence rate [1,2]. While chemotherapy and radiation therapy are applied for PDAC at any stage, tumor resection is essential. Even in resectable cases, PDAC cells may persist and lead to a high rate of recurrence, which necessitates neoadjuvant or adjuvant therapy, including chemotherapy or chemoradiotherapy [3,4]. The poor treatment response of PDAC is a serious clinical problem that warrants novel therapeutic strategies and modalities, including radiosensitization, to improve outcomes in patients with PDAC [5].

PDAC is characterized by extensive fibrosis (desmoplasia) and hypovascularity, leading to poor drug penetration beyond the tumor’s vascularized regions; this unique microenvironment feature of PDAC preferentially contributes to chemoresistance by preventing the perfusion and delivery of chemotherapeutic drugs into the tumors [6,7]. In a study on a genetically engineered mouse model with hypovascular tumors relative to normal tissues, the tumors showed poor drug delivery [8]. Whatcott et al. identified desmoplasia as a poor prognostic factor, further emphasizing the difficulty in achieving effective drug concentrations for treating PDAC and underscoring the urgency for innovative treatment approaches [9].

Radiotherapy with alpha particles (targeted alpha therapy [TAT]) constitutes a promising strategy for cancer therapy [10]. Delivery agents, such as small compounds, peptides, and antibodies labeled with radioisotopes, systemically deliver radioisotopes to tumor tissues to irradiate tumor cells. Labeled drugs are used in the order of picomolar to micromolar concentrations. Unlike those of conventional chemotherapy, the therapeutic effects of TAT depend on radiation dosimetry inside the body rather than the intratumoral drug concentrations [11]. In addition, alpha particles are highly cytotoxic because of their high linear-energy transfer (LET). High LET radiation causes dense ionization within the cell nucleus, resulting in double-stranded DNA breaks (DSBs) and subsequent cell death. Another advantage of a high LET is that ionization is not dependent on the oxygen concentration. Among the therapeutic radionuclides, ^225^Ac is one of the most attractive candidates, as it has a relatively long half-life (9.9 days) and undergoes four alpha and two beta decays to become stable ^209^Bi at 28 MeV. Kratochwil et al. reported that ^225^Ac-PSMA drastically decreased prostate-specific antigen levels and induced a complete response without hematotoxicity [12]. However, the precise mechanism by which ^225^Ac small molecules, including peptides, kill tumor cells is poorly understood except for their high LET and long half-life. Thus, elucidating such mechanisms is critical for optimizing their therapeutic potential in pancreatic cancer and other malignancies.

To enable targeted therapy for PDAC, the α_v_β_3_ and α_v_β_6_ integrins are prominently expressed in PDAC cells [13,14], and both of these integrins recognize arginine–glycine–aspartate (RGD) motifs, making radiolabeled RGD peptides promising candidates for tumor targeting. Radiolabeled c(RGDfK) peptides are also candidate imaging agents for tumors that express α_v_β_3_ integrin, such as glioblastoma and melanoma [15,16,17]. Additionally, ^18^F-FB-A20FMDV2 (NAVPNLRGDLQVLAQKVART), a 20-amino acid peptide that is derived from the foot-and-mouth disease virus, has been utilized as a selective imaging agent of α_v_β_6_ integrin in a pancreatic cancer xenograft model [18]. We previously reported that ^111^In-DOTA-c(RGDfK) specifically accumulates in PDAC in a chemical carcinogenesis model, and we visualized it using single-photon emission computed tomography (SPECT)/computed tomography (CT) [19]. Moreover, ^111^In-DOTA-c(RGDfK) exhibited widespread distribution within tumors with large stroma. Furthermore, c(RGDfK) binds to both α_v_β_6_ integrin and α_v_β_3_ integrin [20], indicating the potential utility of radiolabeled c(RGDfK) peptides as candidates for TAT in PDAC. However, in terms of radiation dosimetry to tumors, dimeric RGD peptides are superior to monomeric RGD peptides such as ^111^In-DOTA-c(RGDfK).

In this study, we aimed to investigate the therapeutic potential of ^225^Ac-DOTA-E[c(RGDfK)]_2_ (^225^Ac-DOTA-RGD_2_) for treating PDAC. The mechanisms underlying the long-lasting therapeutic effects of ^225^Ac small molecules, distinct from antibodies, remain unclear in clinical settings and warrant further investigation. Further, we present the specific cytotoxicity of ^225^Ac-DOTA-RGD_2_ on tumor cells with α_v_β_3_ or α_v_β_6_ integrins and explore the remarkable therapeutic effects in PDAC mice models despite rapid tumor clearance.

## 2. Materials and Methods

### 2.1. Preparation of Radiolabeled DOTA-E[c(RGDfK)]_2_

The radiolabeling of DOTA-RGD_2_ with ^111^In was performed as described previously [21]. Briefly, ^111^InCl_3_ (Nihon Medi-Physics, Tokyo, Japan) and 3 M ammonium acetate (pH 6.0) were added to a microtube and incubated for 5 min. DOTA-E[c(RGDfK)]_2_ (Peptide International Inc., Louisville, KY, USA) was added to this solution and incubated at 95 °C for 10 min. ^111^In-DOTA-E[c(RGDfK)]_2_ was purified using high-performance liquid chromatography.

^225^Ac(NO_3_)_3_ (Oak Ridge National Laboratory, Oak Ridge, TN, USA) was dissolved to 20 MBq/mL in 0.2 M hydrochloric acid (Ultrapure grade HCl; Kanto Chemical Co., Tokyo, Japan) before use. The pH of the ^225^Ac solution was adjusted to approximately 9.0 using 0.2 M Tris buffer (pH 9.0). DOTA-E[c(RGDfK)]_2_ was added to the solution and incubated at 95 °C for 30 min. The radiochemical purity of ^225^Ac-DOTA-RGD_2_ was measured without further purification by thin-layer chromatography using a bioimaging analyzer (Fuji BAS-5000; Fuji Film Co., Tokyo, Japan).

### 2.2. Cell Culture and Animal Model

Human PDAC cell lines AsPC-1, Capan-1, and PANC-1 were purchased from the American Type Culture Collection (Manassas, VA, USA), and BxPC-3 and PSN-1 were purchased from the European Collection of Authenticated Cell Culture (UK Healthy Security Agency, London, UK). AsPC-1, BxPC-3, and PSN-1 cells were cultured in RPMI 1640 supplemented with 10% fetal bovine serum (FBS). Capan-1 cells were cultured in Iscove’s modified Dulbecco’s medium supplemented with 20% FBS, and PANC-1 cells were cultured in Dulbecco’s modified Eagle’s medium supplemented with 10% FBS. All cell lines were grown in a humidified atmosphere of 5% CO_2_ in air at 37 °C. Mycoplasma testing was regularly performed on all cell lines.

Human PDAC xenograft models were used for the experiments. Female BALB/c nude mice (4–5 weeks old, 12–18 g) were obtained from CLEA Japan (Tokyo, Japan). Before the experiments, the mice were acclimatized for ≥1 week. Each PDAC cell line (3–5 × 10^6^ cells) was suspended in a 1:1 ratio of culture media/Matrigel (Corning Inc., Corning, NY, USA) and subcutaneously injected into the right backs of the mice. The study protocol was approved by the Committee for Ethics of Animal Experimentation of the National Cancer Center (K18-012). Animal experiments were performed in accordance with the Guidelines for the Care and Use of Experimental Animals established by the committee.

### 2.3. Cell Viability Assay

The cells were seeded in a 96-well plate and incubated for 24 h (500–4000 cells/well). ^225^Ac-DOTA-RGD_2_, ^225^Ac-DOTA, or phosphate-buffered saline (PBS) as control was added to each well so that the final concentrations were as indicated in Figure 1. After 24 h incubation, the medium was replaced with 100 μL of fresh culture medium. After incubation for 6–10 days, cell viability was assayed using the CellTiter Glo-2.0 luminescent cell viability assay (Promega, Madison, WI, USA). Luminescence activity was measured using SpectraMax iD5 (Molecular Devices, San Jose, CA, USA). IC50 values were calculated using GraphPad Prism 9.3.1 (GraphPad Software, La Jolla, CA, USA). All assays were performed 3–4 times.

### 2.4. FCM Analysis

To estimate the expression of α_v_β_3_, α_v_β_5,_ and α_v_β_6_ integrins, the cells were trypsinized into a single-cell suspension. After washing in PBS, the cells (1 × 10^6^) were resuspended in 100 μL of reaction buffer (1% bovine serum albumin [BSA], 1 mM MgCl_2_, 0.1% NaN_3_, PBS) and incubated with primary antibodies at 4 °C for 30 min. Primary antibodies were detected using the corresponding secondary antibodies. Mouse anti-human α_v_β_3_ integrin antibody (clone LM609, Cat. no. MAB1976; Millipore Sigma, Burlington, MA, USA), mouse anti-human α_v_β_5_ integrin antibody (clone P1F6, Cat. no. MAB1961Z, Millipore Sigma), and mouse anti-human α_v_β_6_ integrin antibody (clone 10D5, Cat. no. MAB2077Z, Millipore Sigma) were used as primary antibodies. FITC rat anti-mouse IgG1 antibody (Cat. no. 553443, BD Biosciences, Franklin Lakes, NJ, USA) and Alexa Fluor 647 donkey anti-mouse IgG antibody were used as secondary antibodies. The cells were analyzed using a FACS aria (BD Biosciences), and the data were analyzed using FlowJo v10 (FlowJo, Ashland, OR, USA).

Staining for γH2AX was conducted as described in previous reports [22,23]. The cells were treated with ^225^Ac-DOTA-RGD_2_ (0, 1, 5, and 10 kBq/mL) for 8, 16, or 24 h. The cells were fixed with 2% paraformaldehyde (PFA) at room temperature for 10 min and washed with PBS after removing the PFA. The cells were resuspended in ice-cold 70% ethanol and kept at −20 °C for 2 h. The cells were washed with BSA-T-PBS (1% BSA and 0.1% Triton-X in PBS) and resuspended in 100 μL BSA-T-PBS containing FITC mouse anti-H2A.X Phospho (Ser139) antibody (clone 2F3, Cat. no. 613404; BioLegend, San Diego, CA, USA) in the dark at 23–26 °C for 1 h and washed with BSA-T-PBS. The cells were simultaneously stained with 100 μL PI/RNase Staining Buffer (BD Biosciences) to analyze the cell cycle phase. Samples were analyzed using a Sony SH800 flow cytometer (Sony Biotechnology, San Jose, CA, USA).

### 2.5. Fluorescence Immunohistochemistry

Cells were treated with ^225^Ac-DOTA-RGD_2_ (0–10 kBq/mL) for 24 h and then fixed with 4% PFA/PBS for 15 min. After permeabilization with 0.1% Triton-X/PBS for 15 min, the cells were blocked with 1% BSA/PBS-T for 1 h and incubated with primary and secondary antibodies for 1 h at room temperature. The nuclei were stained with DAPI. Mouse anti-H2A.X Phospho (Ser139) antibody (Clone:2F3, Cat. no. 613401, 1:500; BioLegend) and rabbit-poly anti-53BP1 antibody (Cat. no. NB100-304SS, 1:800; Novus Biologicals, Centennial, CO, USA) were used as primary antibodies. Alexa Fluor-488 alpaca anti-mouse IgG (Cat. no. 615-545-214, 1:800; Jackson ImmunoResearch Inc., West Grove, PA, USA) and Alexa Fluor-594 donkey anti-rabbit IgG (Cat. no. A11007, 1:500; Thermo Fisher Scientific, Waltham, MA, USA) were used as secondary antibodies. Images were acquired using a BZ-9000 fluorescence microscope (Keyence Corp., Osaka, Japan). The foci of γH2AX and 53BP1 on the images were analyzed using Dynamic Cell Count (Keyence Corp.). The average number of foci per cell was >100 from three independent studies.

### 2.6. In Vitro Combination Treatment

For the combination treatment, BxPC-3 and PANC-1 cells were incubated with ^225^Ac-DOTA-RGD_2_ (0, 5, or 10 kBq/mL) for 4 h. The medium was replaced with a fresh culture medium containing ^225^Ac-DOTA-RGD_2_ (0, 0.1, and 0.5 kBq/mL). After 6 days of incubation, cell viability was measured as described in the section on cell viability assays.

### 2.7. Biodistribution and SPECT/CT Imaging of ^111^In-DOTA-RGD_2_

BxPC-3-bearing mice were injected with 74 kBq of ^111^In-DOTA-RGD_2_ via the tail vein. The mice were sacrificed under anesthesia at the indicated time points, and the tissue samples were excised. The tissue samples were weighed, and radioactivity was determined using a gamma counter (2480 Wizard^2^; PerkinElmer, Waltham, MA, USA). Uptake in the organs was expressed as % ID/g.

SPECT/CT was performed as described previously [19,24]. Briefly, the mice were injected with 7.3 MBq of ^111^In-DOTA-RGD_2_. SPECT/CT was performed using a NanoSPECT/CT scanner (Bioscan Inc., Washington, DC, USA), and the imaging data were processed using dedicated software (Nucline v 2.00, Bioscan Inc.) and VivoQuant 3.5 (Invicro LLC., Boston, MA, USA).

### 2.8. Biodistribution and Alpha Camera Imaging of ^225^Ac-DOTA-RGD_2_

Biodistribution analysis was performed as described previously. BxPC-3-bearing mice were injected with 25 kBq of ^225^Ac-DOTA-RGD_2_ via the tail vein. Radioactivity was counted after >12 h to equilibrate the daughter nuclides using a gamma counter (2480 Wizard^2^). Uptake in the organs was expressed as % ID/g.

The ex vivo imaging of tumors and kidneys was conducted using an ionizing radiation quantum imaging detector alpha camera (QScint Imaging Solutions, Tucson, AZ, USA) [25]. Briefly, BxPC-3-bearing mice were injected via the tail vein with 25–50 kBq/100 μL of ^225^Ac-DOTA-RGD_2_. The mice were sacrificed 1 h after injection, and the tumors were excised. The excised tumors were embedded in the Tissue-Tek O.C.T. compound (Sakura Finetek, Torrance, CA, USA) and rapidly frozen. Cryosections (5–10 μm) were mounted on a ZnS scintillation film (EJ-440; Eljen Technology, Sweetwater, TX, USA) for alpha camera imaging and on glass slides for hematoxylin and eosin staining.

### 2.9. Therapy with ^225^Ac-DOTA-RGD_2_

Mice bearing BxPC-3 tumors were randomized into four groups (*n* = 5–7/group). The mice were intravenously injected with ^225^Ac-DOTA-RGD_2_ (20, 40, or 65 kBq) or saline. During the observation period, the mice were weighed, and the tumor size was measured weekly using a caliper. The tumor volume was calculated as follows: tumor volume (mm^3^) = 1/2 × L × W^2^, where L is the long axis (mm), and W is the width (mm). For the hematotoxicity evaluation, blood was collected from the tail vein at the indicated date. The concentrations of white blood cells, red blood cells, hemoglobin, and platelets were determined using a hematological analyzer (Celltac MEK-6458; Nihon Kohden, Tokyo, Japan). To evaluate kidney injury caused by therapy, urine samples were collected, and neutrophil gelatinase-associated lipocalin (NGAL) was measured using the NGAL ELISA Kit (BioPorto, Hellerup, Denmark). The mice were sacrificed when the tumor volume reached a humane endpoint (weight loss of ≥20% compared with the previous week or tumor volume exceeding 2000 mm^3^). Blood was collected at the end of the study to evaluate alanine aminotransferase (ALT), aspartate transaminase (AST), blood urea nitrogen (BUN), and creatinine. Biochemical parameters were measured using a DRI-CHEM NX700 analyzer (Fujifilm, Tokyo, Japan).

### 2.10. Statistical Analyses

Data were analyzed using GraphPad Prism 9 (GraphPad Software). Differences between groups were assessed using a one-way analysis of variance with Dunn’s multiple-comparison test or a two-way analysis of variance with Dunnett’s test for multiple comparisons. Statistical significance was established at *p* < 0.05 (* *p* < 0.05, ** *p* < 0.01, *** *p* < 0.001, and **** *p* < 0.0001). Kaplan–Meier survival curve statistics were analyzed with a log-rank (Mantel–Cox) test.

## 3. Results

### 3.1. In Vitro Cytotoxicity Assessment by ^225^Ac-DOTA-RGD_2_ Binding

We evaluated the cytotoxicity of ^225^Ac-DOTA-RGD_2_ and the expression of α_v_β_3_, α_v_β_5_, and α_v_β_6_ integrins in PDAC cell lines (Figure 1 and Appendix A). After 24 h of treatment, ^225^Ac-DOTA-RGD_2_ reduced the tumor cell viability in a dose-dependent manner (Figure 1a). The IC_50_ values of ^225^Ac-DOTA-RGD_2_ were 1.16 ± 0.09 kBq/mL for BxPC-3, 2.29 ± 0.69 kBq/mL for Capan-1, 2.41 ± 0.10 kBq/mL for PANC-1, 3.29 ± 0.63 kBq/mL for PSN-1, and 4.72 ± 1.73 kBq/mL for AsPC-1. Meanwhile, the IC_50_ value of ^225^Ac-DOTA for BxPC-3 was 4.17 ± 0.53 kBq/mL (Figure 1b). Flow cytometry (FCM) analysis revealed that these PDAC cells express α_v_β_3_, α_v_β_5,_ and α_v_β_6_ integrins (Appendix A). α_v_β_3_ integrin was abundantly expressed in PANC-1, and α_v_β_6_ integrin was expressed in BxPC-3 and Capan-1. α_v_β_5_ integrin was expressed among all PDAC cell lines at the same level.

### 3.2. γH2AX/53BP1 Expression and G2/M Arrest After ^225^Ac-DOTA-RGD_2_

Fluorescence immunohistochemistry and FCM analysis were performed to detect the induction of DSBs and changes in the cell cycle. The number of γH2AX and 53BP1 foci significantly increased with an increase in the treatment dose of ^225^Ac-DOTA-RGD_2,_ and both numbers were similar (Figure 2a,b). The size of the foci increased with the increase in the treatment dose. The 53BP1 foci were well merged with the γH2AX foci (Figure 2a).

Furthermore, FCM analysis revealed that, based on the duration and dose of ^225^Ac-DOTA-RGD_2_ treatment, γH2AX was upregulated (Figure 3). After 8 h of incubation with 5 and 10 kBq/mL of ^225^Ac-DOTA-RGD_2_, γH2AX expression in BxPC-3 cells was markedly induced (14.4% and 29.0%, respectively; Figure 3a,b); after 24 h of incubation, it reached 2.16% in the control group (0 kBq/mL), 8.03% in the 1 kBq/mL group, 38.39% in the 5 kBq/mL group, and 69.23% in the 10 kBq/mL group. In addition, significantly increased G2/M fractions were observed in the cells treated with 5 and 10 kBq/mL of ^225^Ac-DOTA-RGD_2_ (Figure 3a,c). At an early time point (8 h after incubation), the G2/M fraction in the 10 kBq/mL group increased to 45.4% (vs. 23.72% in controls).

### 3.3. DOTA-RGD_2_ Pharmacokinetics and Intratumoral Distribution

We assessed the pharmacokinetics of ^225^Ac-DOTA-RGD_2_ in BxPC-3 cell-bearing mice (Figure 4a). ^225^Ac-DOTA-RGD_2_ was highly taken up in BxPC-3 (4.80 ± 0.38%ID/g at 1 h) and was gradually washed out. Tumor uptake of ^225^Ac-DOTA-RGD_2_ was still retained up to 1.9 ± 0.63%ID/g at 168 h after injection. In contrast, ^225^Ac-DOTA-RGD_2_ was rapidly removed from the blood (<0.02% injected dose [ID]/g at 4 h). The highest uptake was found in the kidneys (12.87 ± 2.75% ID/g at 15 min), and the radioactivity was rapidly and preferentially excreted in urine (46.47% ID at 24 h and 65.29% ID at 168 h). Radioactivity in the feces was approximately 10%. This pharmacokinetic profile was similar to that of ^111^In-DOTA-RGD_2,_ which exhibited rapid clearance from the blood and high accumulation in the kidneys (Appendix A). The SPECT images clearly showed rapid excretion into the bladder. Moreover, we investigated the effect of the specific activity on the biodistribution of ^111^In-DOTA-RGD_2_ (Appendix A). Co-injection of DOTA-RGD_2_ dose-dependently inhibited the uptake of ^111^In-DOTA-RGD_2_ in BxPC-3 cells and the kidneys. While the administration of 10 μg of DOTA-RGD2 significantly inhibited the uptake by more than 50%, the reduction observed with 1 μg of DOTA-RGD2 was more moderate, with decreases of 33.4% and 31.2% observed in BxPC-3 cells and the kidneys, respectively. Alpha camera imaging and hematoxylin and eosin staining revealed that ^225^Ac-DOTA-RGD_2_ accumulated in tumor areas with higher cellularity (Figure 4b).

### 3.4. Treatment with ^225^Ac-DOTA-RGD_2_ in Mice Bearing BxPC-3

The therapeutic potential and adverse effects of ^225^Ac-DOTA-RGD_2_ were examined in mice bearing BxPC-3 cells. ^225^Ac-DOTA-RGD_2_ inhibited tumor growth in a dose-dependent manner (Figure 5a–c). No significant acute toxicity was observed for the administration of up to 90 kBq, as evaluated by monitoring bodyweight loss > 20%; however, transient body weight loss was observed within 2 weeks (Figure 5d). Low-dose administration (20 and 40 kBq) resulted in a limited inhibition of tumor growth. The inhibitory effects of the two compounds were similar. Doses of 65 and 90 kBq of ^225^Ac-DOTA-RGD_2_ preferentially suppressed tumor growth. Kaplan–Meier analysis revealed that ^225^Ac-DOTA-RGD_2_ significantly increased the median survival time compared with that in non-treated animals (control: 63 days, 20 kBq: 70 days, *p* < 0.05; 40 kBq: 84 days, *p* < 0.001; 65 kBq: 112 days, *p* < 0.001; 90 kBq: and 126 days, *p* < 0.001; Figure 5c). Hematological analysis revealed that the white blood cell counts transiently decreased and later recovered readily (Appendix A). Similarly, platelet count decreased after the administration of ^225^Ac-DOTA-RGD_2_. In particular, 65 and 90 kBq of ^225^Ac-DOTA-RGD_2_ induced a long-lasting decrease in platelet count. Furthermore, we measured neutrophil gelatinase-associated lipocalin (NGAL) levels to evaluate acute kidney injury at 3, 7, and 14 days after the administration of ^225^Ac-DOTA-RGD_2_. There was no significant increase in NGAL by ^225^Ac-DOTA-RGD_2_ (Appendix A). Alanine aminotransferase (ALT), aspartate transaminase (AST), blood urea nitrogen (BUN), and creatinine levels were measured at the endpoint. ^225^Ac-DOTA-RGD_2_ did not increase the ALT, AST, BUN, or creatinine levels (Appendix A).

### 3.5. In Vitro Antitumor Activity of ^225^Ac-DOTA-RGD_2_ Combination

We investigated the effects of low-dose radiation treatment following transient high-dose radiation treatment on cell viability (Figure 6). Six days of incubation with 0.1 kBq/mL of ^225^Ac-DOTA-RGD_2_ did not affect cell viability (Figure 6a). The effect of 0.5 kBq/mL of ^225^Ac-DOTA-RGD_2_ was moderate. However, only the 4 h incubation with 5 or 10 kBq/mL of ^225^Ac-DOTA-RGD_2_ drastically enhanced the antitumor activity of 0.5 kBq/mL of ^225^Ac-DOTA-RGD_2_. In BxPC-3 cells, the cell viability following the combination treatment was 42.7 ± 10.8% for 5–0.5 kBq/mL and 29.5% ± 4.8% for 10–0.5 kBq/mL, whereas that after the transient treatment of 5 and 10 kBq/mL was 86.8% ± 6.8% and 65.4% ± 5.0%, respectively. In PANC-1 cells, the cell viability after the combination treatment was 62.9% ± 11.8% for 5–0.5 kBq/mL and 48.1% ± 8.1% for 10–0.5 kBq/mL, while that following the transient treatment of 5 and 10 kBq/mL was 95.7% ± 5.1% and 79.5% ± 4.7%, respectively.

The 4 h incubation period with 5 or 10 kBq/mL of ^225^Ac-DOTA-RGD_2_ did not alter the cell cycle distribution, although following a subsequent 16 h incubation with 0.1 or 0.5 kBq/mL of ^225^Ac-DOTA-RGD_2_ (totaling 24 h), there was an increase in the G2/M fraction (Figure 6b). By the 48 h mark, the cell cycle distribution had returned to levels comparable to the control, characterized by a predominance of the G1 fraction.

γH2AX expression was notably induced by incubation with 0.5 kBq/mL of ^225^Ac-DOTA-RGD_2_. In the absence of transient incubation with 5 or 10 kBq/mL of ^225^Ac-DOTA-RGD_2_, only 0.5 kBq/mL of the compound increased γH2AX expression (7.47% ± 1.82% for 0–0.5 kBq vs. 2.30% ± 0.82% for 0 kBq and 3.98% ± 1.91% for 0–0.1 kBq at 48 h). The 4 h incubation with 5 or 10 kBq/mL of ^225^Ac-DOTA-RGD_2_ significantly boosted γH2AX expression induced by 0.5 kBq/mL of ^225^Ac-DOTA-RGD_2_ and 0.1 kBq/mL of ^225^Ac-DOTA-RGD_2_. Meanwhile, the γH2AX expression induced by 5 or 10 kBq/mL of ^225^Ac-DOTA-RGD_2_ decreased over time.

## 4. Discussion

A single administration of ^225^Ac radiopharmaceuticals confers significant antitumor efficacy in preclinical and clinical studies. The first clinical study using ^225^Ac-PSMA-617 reported a drastic positive response even after a single administration, and finally, a complete response was achieved after three administrations [12]. In a preclinical study, ^225^Ac-DOTATATE delayed tumor growth in a lung neuroendocrine tumor model [26]. However, the mechanisms underlying these potent antitumor effects remain obscure. In general, alpha nuclides have been focused on for high LET, whereas the long half-life of ^225^Ac makes it one of the most attractive candidates for tumor treatment. Here, we performed efficacy and toxicity studies and investigated the mechanism underlying the continuous antitumor effect of ^225^Ac radiopharmaceuticals in a pancreatic tumor model using ^225^Ac-DOTA-RGD_2_.

Our study indicated that ^225^Ac-DOTA-RGD_2_ would be a possible candidate for TAT for pancreatic cancer with α_v_β_3_/α_v_β_6_ integrins. In vitro cytotoxicity experiments indicated that ^225^Ac-DOTA-RGD_2_ was more potent than ^225^Ac-DOTA in BxPC-3 cells. In addition, a significant strong correlation was observed between the in vitro cytotoxicity (IC_50_) and tumor uptake of ^225^Ac-DOTA-RGD_2_ (Appendix A; r = −0.906, *p* < 0.05). Therefore, the cytotoxicity of ^225^Ac was due to the binding of ^225^Ac-DOTA-RGD_2_ to tumor cells. However, 24 h of incubation with 10 kBq/mL of ^225^Ac-DOTA, which does not bind to α_v_β_3_/α_v_β_6_ integrins, was also lethal to the cells. This implies that radiation dosimetry to non-target tissues should be well managed in TAT, although the range of alpha rays is <100 μm.

This preclinical study revealed that ^225^Ac-DOTA-RGD_2_ (20–90 kBq), with a tumor accumulation of 2.0% ID/g after 48 h, significantly suppressed tumor growth. Our previous study using ^90^Y-DOTA-c(RGDfK) indicated that even with three injections of 11.1 MBq, tumor growth inhibition was limited [21]. A single high-dose injection of ^90^Y-DOTA-RGD_2_ (37 MBq) delayed tumor growth (median survival of 54 days vs. 19 days in the untreated group) in the OVCAR-3 xenograft model, although regrowth was observed 2 weeks after treatment. The prolongation of median survival time would be caused by the high LET and long half-time (9.9 days vs. 64 h of ^90^Y) of ^225^Ac. ^225^Ac-labeled antibodies have been reported to be more effective in tumor treatment than antibodies labeled with a beta radionuclide (^90^Y) or a short-half-life alpha nuclide (^213^Bi) [27,28]. Thus, our therapeutic experiments using ^225^Ac-DOTA-RGD_2_ suggest that ^225^Ac-labeled peptides could confer drastic therapeutic efficacy, although specific tumor-targeting ability is required.

Regarding side effects, kidney injury is debatable. Mice treated with 111 kBq of ^225^Ac-DOTATATE showed chronic nephropathy, although BUN and creatinine levels did not increase [26]. This side effect is characterized by pharmacokinetics unique to peptides that show preferential renal excretion [29]. However, in our study, ^225^Ac-DOTA-RGD_2_ was significantly taken up by the kidneys, although renal dysfunction was not detected by biochemical analysis. Additionally, TAT using ^225^Ac-DOTATATE has been attempted in patients with neuroendocrine tumors [30]. Yadav et al. reported that ^225^Ac-DOTATATE treatment is effective without severe hematological, renal, and hepatological toxicities and is beneficial to patients refractory to ^177^Lu treatment [31]. Dose planning based on dosimetry from preclinical studies is required to avoid radiation nephrotoxicity.

Considering the potent antitumor activity of ^225^Ac-DOTA-RGD_2_, we should focus on the fact that the cells were efficiently blocked at G2/M by ^225^Ac irradiation. FCM analysis revealed that treatment with >5 kBq/mL ^225^Ac-DOTA-RGD_2_ readily led cells to G2/M arrest, and γH2AX was sequentially found in the cells. Irradiation with high-LET carbon ions (^12^C^6+^) at 2 Gy is more lethal and induces G2/M arrest and γH2AX expression more efficiently compared with X-ray irradiation at the same dose [32]. It is well known that cells in the G2/M phase are sensitive to radiation [33]. Some radiosensitizing agents arrest cells in the G2/M phase [34,35,36]. Thus, targeting the cell cycle is a strategy to enhance radiosensitivity [37]. This may allow a lower radiation dose to severely damage the cells. Rapid clearance of peptides or peptide mimetics from normal organs enables the administration of high radioactivity to enhance radiosensitivity.

From our results and reports on ^225^Ac-PRRT thus far, we speculate that, after transient intense irradiation, continuous low-dose irradiation would contribute to successful therapeutic efficacy. Based on our in vitro results showing that G2/M arrest was induced by ^225^Ac-DOTA-RGD_2_, we examined the effect of combination treatment with ^225^Ac-DOTA-RGD_2_ on BxPC-3 and PANC-1 cells. Pretreatment with 5 or 10 kBq/mL of ^225^Ac-DOTA-RGD_2_ for only 4 h significantly enhanced the cytotoxicity of low-dose ^225^Ac-DOTA-RGD_2_ (0.1 or 0.5 kBq/mL). This suggests that an initial higher dose triggers sufficient cytotoxicity, even at subsequent lower doses. These results would explain the continuous suppression of tumor growth by ^225^Ac-DOTA-RGD_2_. Therefore, the pharmacokinetics of the peptides are suitable for TAT using radionuclides with long half-lives.

In addition to G2/M arrest, alterations in the DNA repair response could be another factor that induces radiosensitization. Generally, DNA damage is repaired by homologous recombination (HR) and non-homologous end joining (NHEJ). Severe DNA damage, such as clustered DSBs in heterochromatin, is repaired by HR following NHEJ at an early stage [38]. HR functions in the late S/G2 phase using a sister chromatid as a template to repair DSBs precisely [39]. Foci of the DNA damage response mediator 53BP1 were well merged with γH2AX foci in BxPC-3 and PANC-1 cells during 24 h treatment with ^225^Ac-DOTA-RGD_2_. 53BP1 is a key factor in regulating the balance between NHEJ and HR. 53BP1 suppresses HR and facilitates NHEJ, a fast repair process that occurs within a few hours of DNA damage [40]. The inhibition of HR by the Chk1/2 inhibitor AZD7762 induced radiosensitization and a prolonged tumor-volume doubling time in MiaPaCa-2 and patient-derived pancreatic tumor xenograft models [41]. Sustainable irradiation with ^225^Ac-DOTA-RGD_2_ maintains 53BP1 expression, resulting in the inhibition of HR.

In TAT, it is controversial whether antibodies or peptides are preferable. Antibodies that exhibit slow blood clearance and sustained tumor accumulation are potential candidates as ^225^Ac radiopharmaceuticals. Theoretically, antibodies labeled with long-half-life radionuclides, such as ^225^Ac and ^177^Lu, can continuously irradiate tumors. With the trend of antibody–drug development, labeled antibodies have been vigorously studied [42]. Unfortunately, only ^90^Y-labeled anti-CD20 antibody (Zevalin^®^) is currently approved as an antitumor drug. Slow blood clearance and non-specific intense uptake in the liver may result in hematotoxicity and liver toxicity, respectively [43,44,45]. Therefore, a lower dose is required to reduce side effects. In the case of peptides, the radioactivity per administration should be increased because they exhibit relatively rapid clearance. In addition, multiple doses are available (or mandatory) for adequate therapeutic effects while controlling side effects. ^225^Ac-PSMA-617 and ^117^Lu-DOTATATE (Lutathera^®^) are administrated multiple times [30,46].

Additionally, peptides are superior to antibodies in terms of intratumoral permeability. We previously reported that ^111^In-DOTA-c(RGDfK) was homogeneously distributed inside pancreatic cancer cells in a carcinogenesis model that was histologically similar to human pancreatic cancer [19]. Furthermore, alpha camera imaging confirmed the uniform distribution of ^225^Ac-DOTA-RGD_2_ in the viable region of BxPC-3 cells. However, radiolabeled antibodies accumulate in the rim, which corresponds to viable tumor tissue and areas with high blood vessel density and large blood vessels [47,48]. These features of radiolabeled peptides suggest that they are appropriate for treating PDAC.

This study has a limitation in that it has not demonstrated that combination treatment with high-dose administration followed by a low dose is effective. We need to investigate the dose and dosage interval and demonstrate the therapeutic effects and side effects of the dosing design. Unfortunately, however, the supply of ^225^Ac is extremely limited, and it is extremely difficult to obtain ^225^Ac for basic research. In fact, we currently cannot obtain it from overseas. The production of ^225^Ac is currently progressing in Japan, and we will examine effective dosing designs for clinical use in the future.

In conclusion, high-intensity irradiation with ^225^Ac-DOTA-RGD_2_ induced G2/M arrest, thereby inducing radiosensitivity. Our findings suggest that ^225^Ac-DOTA-RGD_2_ could confer a radiation dose sufficient to treat PDAC without undesirable radiation exposure to normal tissues. The high LET and long half-life of ^225^Ac make it suitable for the development of radionuclide therapies.

## Figures and Tables

**Figure 1 pharmaceutics-17-00009-f001:**
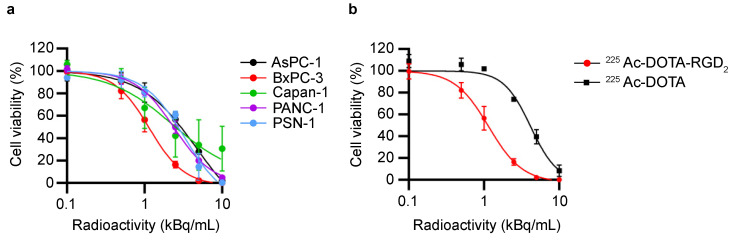
In vitro cytotoxicity. (**a**) Cytotoxicity of ^225^Ac-DOTA-RGD_2_ in human pancreatic tumor cell lines. (**b**) Comparison of cytotoxicity between ^225^Ac-DOTA-RGD_2_ and ^225^AcDOTA in BxPC-3. All assays were performed in triplicate. Data are presented as mean ± standard deviation.

**Figure 2 pharmaceutics-17-00009-f002:**
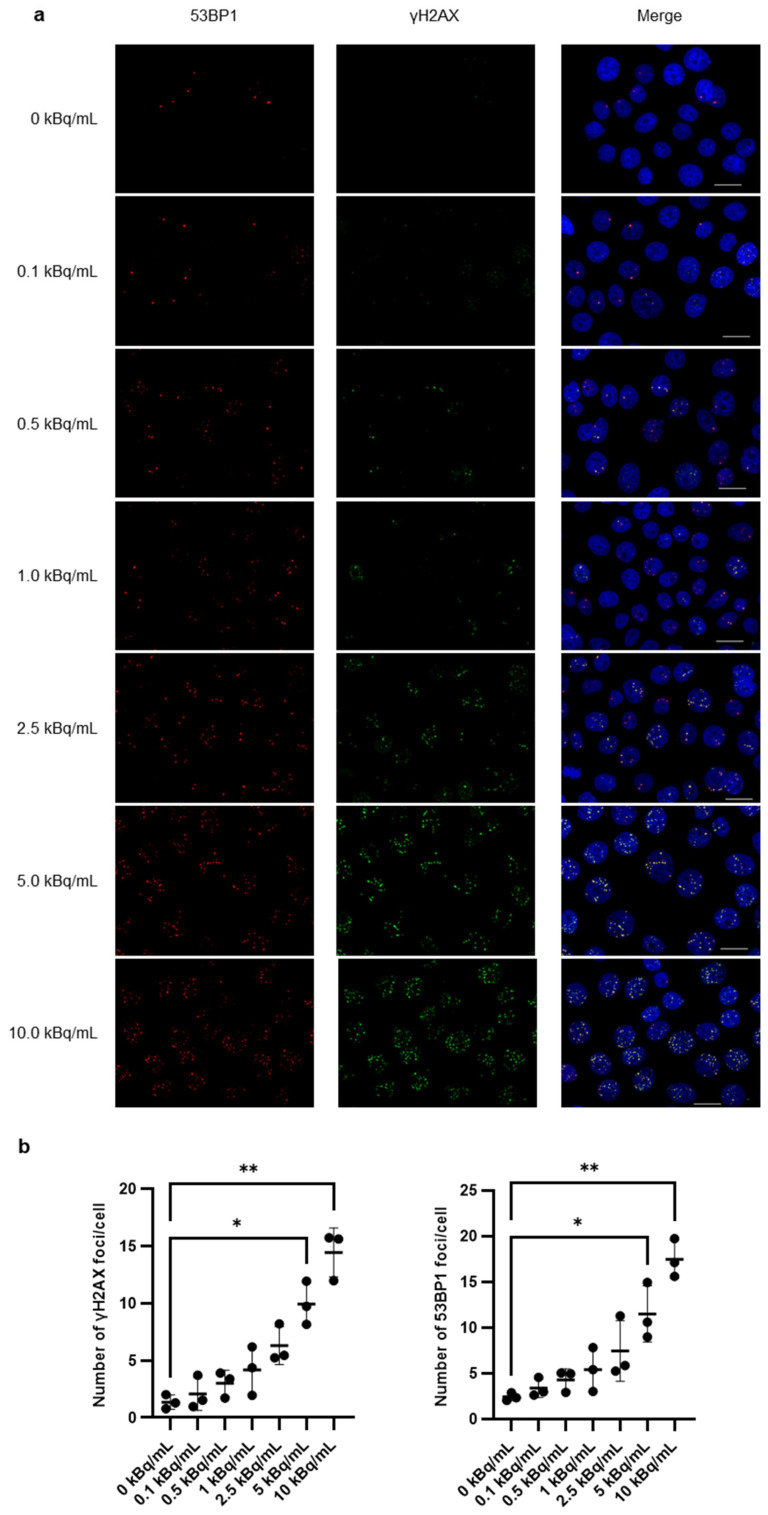
Induction of γH2AX and 53BP1 foci formation in response to increasing doses of ^225^Ac-DOTA-RGD_2_ at 24 h. (**a**) Representative images of γH2AX and 53BP1 foci obtained by immunofluorescence microscopy in BxPC-3 cells. Scale bar, 20 μm. (**b**) The number of γH2AX and 53BP1 foci per cell. Induction of γH2AX and 53BP1 foci in response to increasing doses of ^225^Ac-DOTA-RGD_2_ was monitored at 24 h. The number of γH2AX and 53BP1 foci per cell was counted, and 50–100 cells were analyzed. All assays were performed in triplicate. Data are presented as mean ± standard deviation and analyzed using a one-way analysis of variance with Dunn’s multiple-comparisons test (* *p* < 0.05, ** *p* < 0.01).

**Figure 3 pharmaceutics-17-00009-f003:**
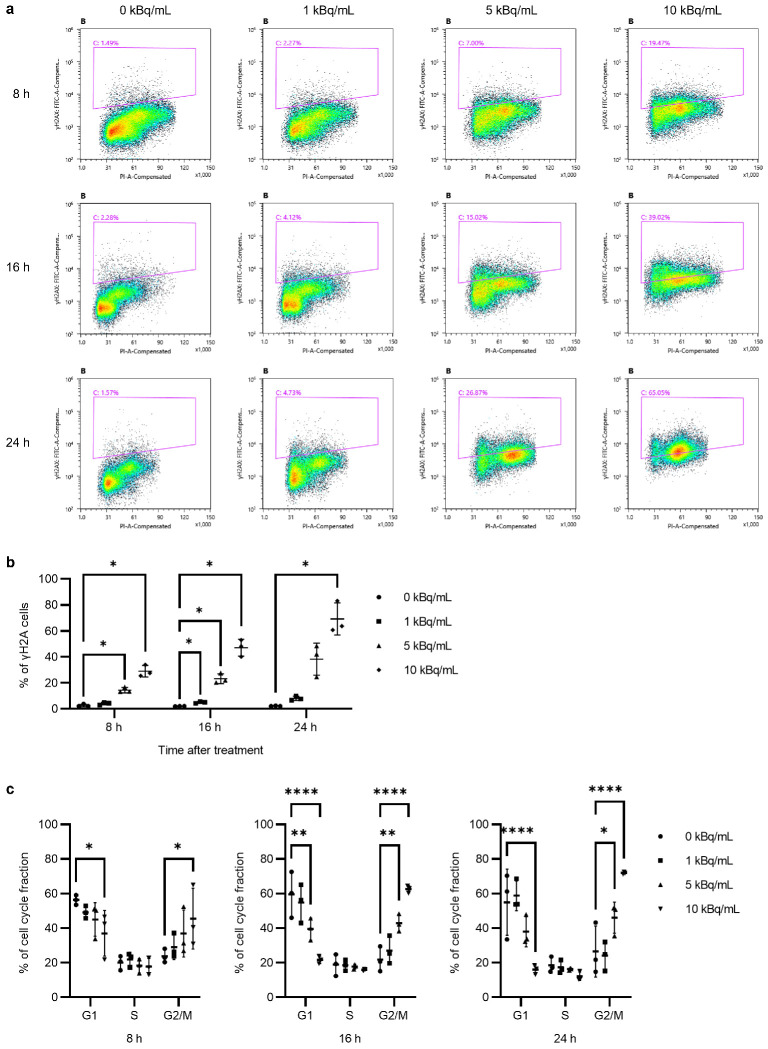
Flow cytometric analysis of BxPC-3 after incubation with ^225^Ac-DOTA-RGD_2_. (**a**) Representative fluorescence-activated cell sorting plots for γH2AX. The *y*-axis indicates γH2AX staining, and the *x*-axis is the DNA content. (**b**) Percentage of cells with γH2AX staining. All assays were performed in triplicate. (**c**) Percentage of cell cycle distribution (G1, S, and G2/M). All assays were performed in triplicate. Data are presented as the mean ± standard deviation (* *p* < 0.05, ** *p* < 0.01, and **** *p* < 0.0001).

**Figure 4 pharmaceutics-17-00009-f004:**
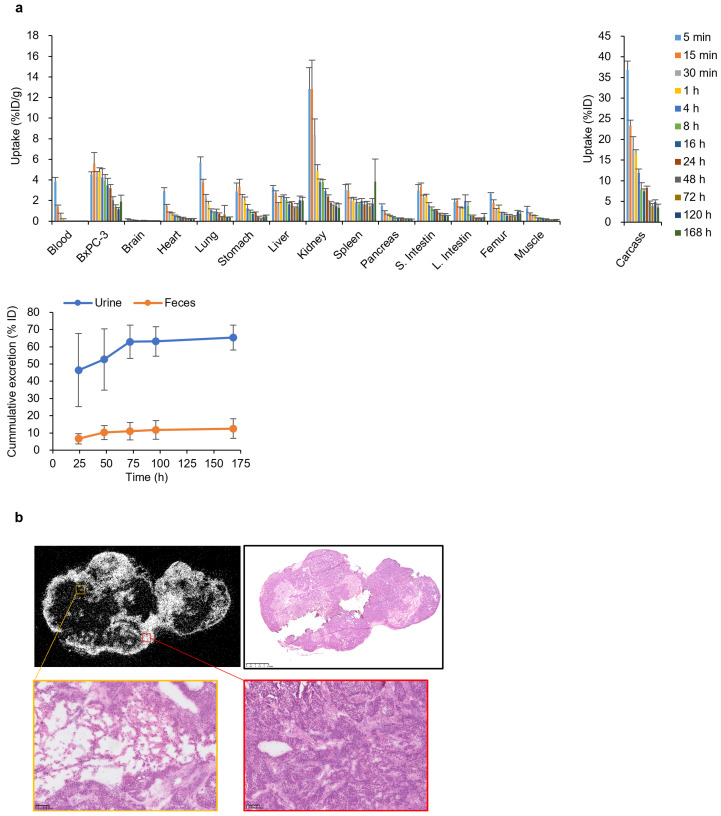
Biodistribution of ^225^Ac-DOTA-RGD_2_ in BxPC-3-bearing mice. (**a**) Pharmacokinetics of ^225^Ac-DOTA-RGD_2_. Data are expressed as % ID/g for organs and blood and as % ID for carcass, urine, and feces. Data are shown as the mean ± standard deviation (*n* = 3–4). (**b**) Alpha camera imaging of intratumoral distribution and corresponding hematoxylin and eosin images. The scale bars indicate 100 μm. ID, injected dose.

**Figure 5 pharmaceutics-17-00009-f005:**
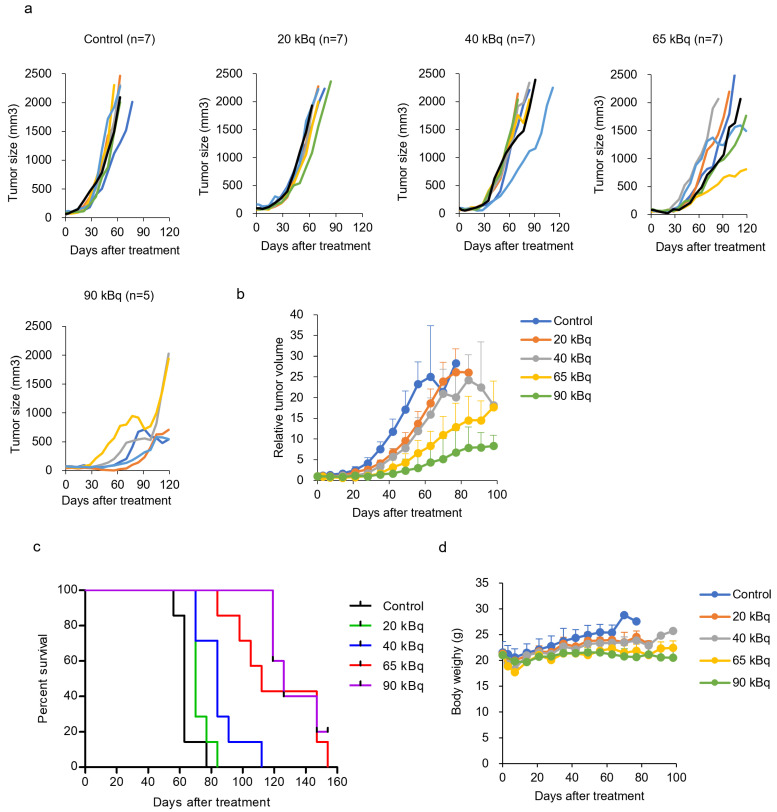
Therapeutic efficacy of ^225^Ac-DOTA-RGD_2_ in BxPC-3-bearing mice. (**a**) Individual tumor responses. Each solid color line represents a tumor from a single mouse. (**b**) Relative tumor growth of the mice groups treated with a single dose of ^225^Ac-DOTA-RGD_2_ compared to the control group (untreated). Data are shown as the mean ± standard deviation. (**c**) Kaplan–Meier survival curves of the mice treated with ^225^Ac-DOTA-RGD_2_. Log-rank (Mantel–Cox) test; *p* = 0.0192, hazard ratio [HR] 2.415, 95% CI 0.7639–7.636 (control vs. 20 kBq); *p* = 0.0014, HR 3.342, 95% CI 0.9631–11.60 (control vs. 40 kBq); *p* = 0.0002, HR 3.774, 95% CI 1.042–13.67 (control vs. 65 kBq); *p* = 0.0009, HR 3.786, 95% CI 1.062–13.49 (control vs. 90 kBq). (**d**) Change in body weight after administration of ^225^Ac-DOTA-RGD_2_. Data are shown as the mean ± standard deviation.

**Figure 6 pharmaceutics-17-00009-f006:**
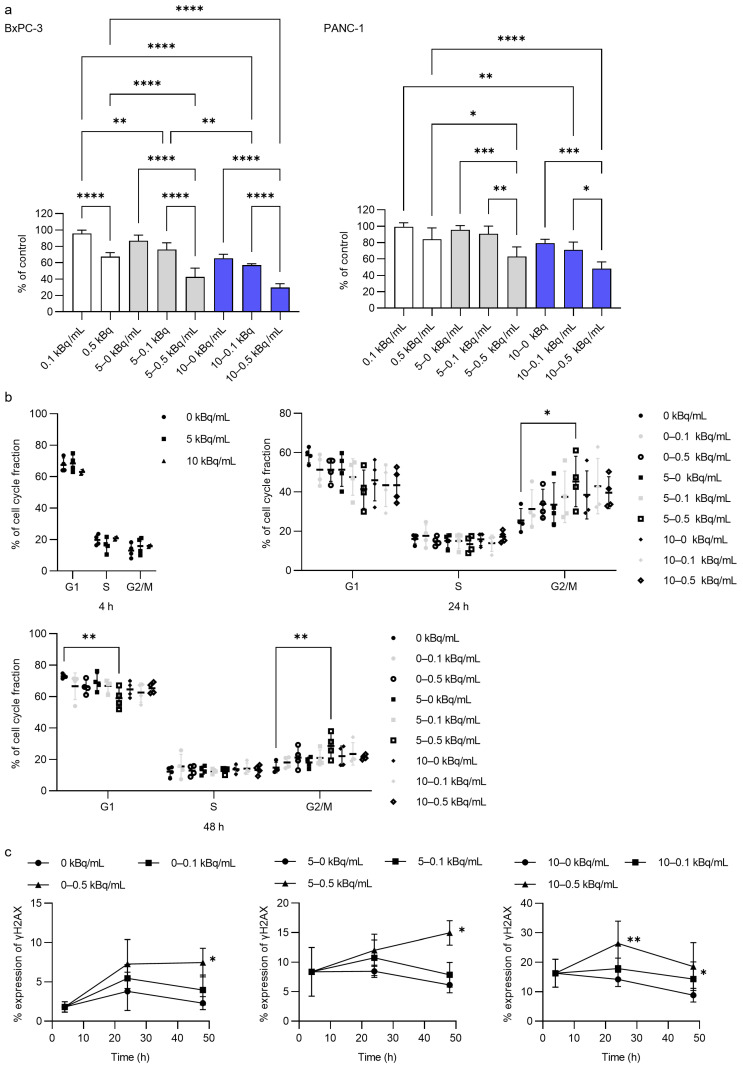
Cytotoxicity, cell cycle, and γH2AX expression by low-dose ^225^Ac-DOTA-RGD_2_ after 4 h of treatment with high-dose (5 or 10 kBq/mL) of ^225^Ac-DOTA-RGD_2_ in BxPC-3 and PANC-1 cells. (**a**) Cell viability. The white, grey, and blue columns indicate the pretreatment with 0, 5, and 10 kBq/mL of ^225^Ac-DOTA-RGD_2_, respectively. (**b**) Percentage of cell cycle distribution. (**c**) Time course of γH2AX expression. The significance of γH2AX expression at each time point was compared to 0, 5, or 10 kBq/mL as the control in each graph. Data represent the mean ± standard deviation (*n* = 2–4). * *p* < 0.05, ** *p* < 0.01, *** *p* < 0.001, and **** *p* < 0.0001.

## Data Availability

The data in this manuscript are available from the corresponding author upon reasonable request.

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
