# Peer review of "Long-Term Therapeutic Effects of 225Ac-DOTA-E[c(RGDfK)]2 Induced by Radiosensitization via G2/M Arrest in Pancreatic Ductal Adenocarcinoma"

_pharmaceutics, 2024, doi:10.3390/pharmaceutics17010009_

Round 1

Reviewer 1 Report

Comments and Suggestions for Authors

The manuscript by Yoshimoto demonstrates that short-term exposure to a high dose of 225Ac-DOTA-RGD2 induces G2/M cell cycle arrest and DNA damage (marked by γH2AX and 53BP1 foci), rendering cells more sensitive to subsequent low-level alpha irradiation ("radiosensitization"). The findings elegantly demonstrate that short-term high-dose exposure enhances the effectiveness of subsequent low-dose treatment in vitro. However, the lack of in vivo exploration of this combination strategy leaves an important gap that warrants further investigation.

Author Response

Comments 1: [The manuscript by Yoshimoto demonstrates that short-term exposure to a high dose of 225Ac-DOTA-RGD2 induces G2/M cell cycle arrest and DNA damage (marked by γH2AX and 53BP1 foci), rendering cells more sensitive to subsequent low-level alpha irradiation ("radiosensitization"). The findings elegantly demonstrate that short-term high-dose exposure enhances the effectiveness of subsequent low-dose treatment in vitro. However, the lack of in vivo exploration of this combination strategy leaves an important gap that warrants further investigation.]

Response 1: Thank you for your valuable comments. I agree with your comment which says that we need further in vivo investigation of the combination strategy. Unfortunately, now the supply of 225Ac has been limited and we can’t investigate it soon. We surely understand this is the limitation of our study. Therefore, we have already described our limitations in the discussion section (p16, last paragraph) as follows. Please confirm if it is enough or not.

This study has a limitation in that it has not demonstrated that combination treatment with high dose administration followed by low dose is effective. we need to investigate the dose and dosage interval and demonstrate the therapeutic effects and side effects of the dosing design. Unfortunately, however, the supply of 225Ac is extremely limited and it is extremely difficult to obtain 225Ac for basic research. In fact, we cannot obtain it from overseas now. Production of 225Ac is currently progressing in Japan up to now, and we will examine effective dosing designs for clinical use in the future.

Reviewer 2 Report

Comments and Suggestions for Authors

The paper is very well written and has adequate experimental results. In conclusion the paper is accepted.

Comments on the Quality of English Language

The quality is good.

Author Response

Comments 1: The paper is very well written and has adequate experimental results. In conclusion the paper is accepted.

Response 1: Thank you for your comment. We are very happy with your decision.

Reviewer 3 Report

Comments and Suggestions for Authors

The paper titled  Long-term therapeutic effects of 225Ac-DOTA-E[c(RGDfK)]2 induced by radiosensitization via G2/M arrest in PDAC” by  Mitsuyoshi Yoshimoto, Kohshin Washiyama, Kazunobu Ohnuki, Ayano Doi, Miki Inokuchi, Motohiro Kojima, Brian W. Miller, Yukie Yoshii, Anri Inaki, Hirofumi Fujii described development and experimental applicaton of alpha-therapeutic  225Ac-DOTA-18 RGD2 developped to target of integrin-expressed tumors.

The paper is very well designed, performed and described. All the data are clearly presented and analyzed. The therapeutic outcome, even not perfect, is sufficiently good to support further research and development of this approach.  A repeated therapy schedule could be considered in future.

Few minor comments are as follows:

-        225Ac-DOTA-RGD2 highly accumulated in BxPC-3 (4.80 ± 0.38%ID/g at 1 h)” , line 295. In the data, there is no accumulation in BxPC-3. Accumulation means that the level increases its value during then time, which is not the case here. A better-describing statement would be “the level was still high…” etc.

-        Fig 5. The control mice start to die about 60 days after the therapy start. It may be sufficient to show the statistical analysis (panel b) only till day 60, where all the mice are alive, together with the comparison of the significance of the different doses compared to control.

-        The individual tumors growth may better go to Supplementary. The averages curve (panel b) is sufficient here.

-        In the discussion (line 419), a nephropathy was mentioned. It can be expected and correlated with the data of tissue distribution, where kidneys are most exposed organs. An appropriate comment would be welcome!

-        Fig. S2b may better serve in main text than in supplementary.

Author Response

Thank you for your valuable comments. As you mentioned, we’d like to investigate a repeated strategy which we are proposing in this study.

Comments 1: [   “225Ac-DOTA-RGD2 highly accumulated in BxPC-3 (4.80 ± 0.38%ID/g at 1 h)” , line 295. In the data, there is no accumulation in BxPC-3. Accumulation means that the level increases its value during then time, which is not the case here. A better-describing statement would be “the level was still high…” etc.]

Response 1: According to your suggestion, we changed the description as follows.

225Ac-DOTA-RGD2 was highly taken up in BxPC-3 (4.80 ± 0.38%ID/g at 1 h) and was gradually washed out.

Comments 2: [ Fig 5. The control mice start to die about 60 days after the therapy start. It may be sufficient to show the statistical analysis (panel b) only till day 60, where all the mice are alive, together with the comparison of the significance of the different doses compared to control.]

Response 2:

I greatfully agree with your comment. As you pointed out, the data up to 60 days may be enough to show the therapeutic effect. However, I believed it would be more useful for readers to show the long-term therapeutic effect and side effects. In addition, the endpoint of this study was set to a tumor volume of 2000 cm2 or more, so I thought it would be a bad idea to delete the data intentionally.

Comments 3: [The individual tumors growth may better go to Supplementary. The averages curve (panel b) is sufficient here.]

Response 3: Thank you for your advice. Panel a means individual tumor volume in mm3, but panel b means relative tumor volume. So they are different. I’d like to show the panel a in the main text, not Supplementary if MDPI permits.

Comments 4: [In the discussion (line 419), a nephropathy was mentioned. It can be expected and correlated with the data of tissue distribution, where kidneys are most exposed organs. An appropriate comment would be welcome!]

Response 4: According to your comment, I added some sentences.

Regarding side effects, kidney injury is controversial. Mice treated with 111 kBq of 225Ac-DOTATATE showed chronic nephropathy, although BUN and creatinine did not increase [26]. This side effect is characterized by pharmacokinetics unique to peptides that show preferential renal excretion [29]. In our study, on the other hand, 225Ac-DOTA-RGD2 was significantly taken up by the kidneys, although renal dysfunction was not detected by biochemical analysis. In addition, TAT using 225Ac-DOTATATE has been attempted in patients with neuroendocrine tumors [30]. Yadav et al. reported that 225Ac-DOTATATE treatment is effective without severe hematological, renal, and hepatological toxicities and is beneficial to patients refractory to 177Lu treatment [31]. Dose planning based on dosimetry from preclinical studies would be required to avoid radiation nephrotoxicity.

Comments 5: [Fig. S2b may better serve in main text than in supplementary.]

Response 5: Thank you for your kind comment. I want Fig. S2b to still be in supplementary because the main topic of this manuscript is the radionuclide therapy using 225Ac-DOTA-RGD2. I think the pharmacokinetics data of 111In-DOTA-RGD2 would be appropriate to be in the supplementary.

Reviewer 4 Report

Comments and Suggestions for Authors

The manuscript by Yoshimoto et al is well written with the experiments well described and thorough in this evolving field of radionuclide therapy using a hard to source yet interesting therapeutic isotope (Actinium 225).

Although there was only one animal study conducted with this study due to the limited supply of the alpha emitting isotope and one using Indium111. The invitro work provides useful information.

Recommend the authors add the Indium 111 animal study into the main paper and not the supplementary section as this can be confusing to follow and will more easily allow comparisons to be made. One cell line used for this manuscript is also a limiting. 

Also suggest that an Isotype control be used as the comparison in this work both in vitro and in vivo to address any non-specific targeting. This may be challenging due to the limited current worldwide availability of Actinium-225.

Author Response

Thank you for your valuable comments. As you mentioned, the supply of 225Ac is very limited. That’s why the study protocol is also limited. However, we believe that this research gives useful information to the readers.

Comments 1: [Recommend the authors add the Indium 111 animal study into the main paper and not the supplementary section as this can be confusing to follow and will more easily allow comparisons to be made. One cell line used for this manuscript is also a limiting.]

Response 1: Thank you for your suggestion. I want Fig. S2 to still be in the supplementary section because the main topic of this manuscript is radionuclide therapy using 225Ac-DOTA-RGD2. The pharmacokinetics of 225Ac-DOTA-RGD2 is more important and the data of 111In-DOTA-RGD2 is just supportive. I think the pharmacokinetics data of 111In-DOTA-RGD2 would be appropriate to be in the supplementary section.

As you mentioned, We used only the BxPC-3 cell line in the therapeutic experiment. However, we used five cell lines in the some in vitro studies (Figurer 1 and S1) and the biodistribution study (Figure S4).

Comments 2: [Also suggest that an Isotype control be used as the comparison in this work both in vitro and in vivo to address any non-specific targeting. This may be challenging due to the limited current worldwide availability of Actinium-225.]

Response 2: Thank you for your comment. As you mentioned, the availability of 225Ac is very limited, so all experiments that we want can not be done. We’d like to do further studies in the future.